# MoBA: Mixture of Bi-directional Adapter for Multi-modal Sarcasm Detection

Yifeng Xie
Guangdong University of Technology
Guangzhou, China
evfxie@gmail.com

Zhihong Zhu*
Peking University
Shenzhen, China
zhihongzhu@stu.pku.edu.cn

Xin Chen
Guangdong University of Technology
Guangzhou, China
xinchen723@126.com

Zhanpeng Chen
Peking University
Shenzhen, China
troychen927@pku.edu.cn

Zhiqi Huang*
Peking University
Shenzhen, China
zhiqihuang@pku.edu.cn

## Abstract

In the field of multi-modal learning, model parameters are typically large, necessitating the use of parameter-efficient fine-tuning (PEFT) techniques. These methods have been pivotal in enhancing training efficiency for downstream tasks in almost all situations. However, directly applying PEFT methods struggles to fully address the intricate demands of multi-modal tasks, such as multi-modal sarcasm detection (MSD), which demands the extraction and comparison of cues from different modalities. MSD, particularly when reliant on textual and visual modalities, faces challenges in identifying sarcasm's incongruity. This issue often arises from the lack of intermodality interaction during tuning, resulting in a disconnect between textual and visual information. In this paper, we introduce a novel approach called **B**i-directional **A**dapter (BA), designated as **MoBA**. This approach is designed to minimize training parameters while enhancing the model's ability to interpret sarcasm across modalities. By facilitating an exchange between textual and visual information through a low-rank representation, our method adeptly captures the nuances of sarcastic expressions with a reduced number of training parameters. Our empirical studies, carried out on two publicly accessible and emerging datasets, demonstrate that our model substantially improves sarcasm detection accuracy. These findings indicate that our approach provides a more reliable and efficient solution to address the complexities of MSD.

## CCS Concepts

• **Information systems → Multimedia information systems**; *Information retrieval*; • **Computing methodologies → Artificial intelligence**.

## Keywords

multi-modal learning, parameter-efficient tuning, mixture of experts

**ACM Reference Format:**
Yifeng Xie, Zhihong Zhu, Xin Chen, Zhanpeng Chen, and Zhiqi Huang. 2024. MoBA: Mixture of Bi-directional Adapter for Multi-modal Sarcasm Detection. In *Proceedings of the 32nd ACM International Conference on Multimedia (MM '24), October 28–November 1, 2024, Melbourne, VIC, Australia.* ACM, New York, NY, USA, 9 pages. https://doi.org/10.1145/3664647.3680914

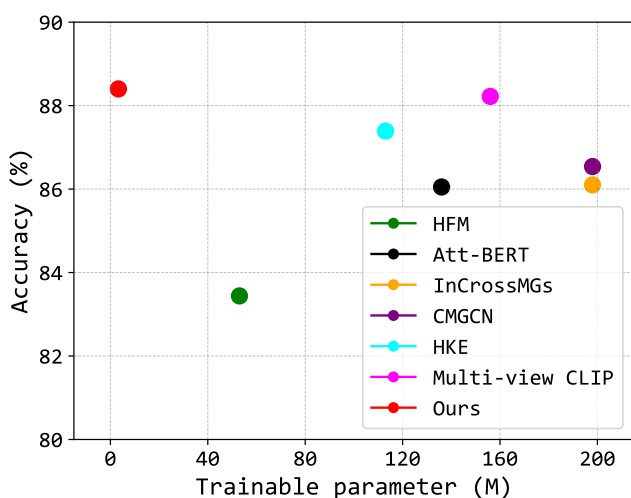

**Figure 1: Comparison of our model with the other exist multimodal methods on the multi-modal sarcasm detection task.**

## 1 Introduction

In the rapidly evolving landscape of artificial intelligence (AI), of artificial intelligence, the strategy of pretraining models has emerged as a critical foundation for developing specialized AI capabilities. This approach has achieved significant success in natural language processing (NLP) [40, 44, 57, 58], computer vision (CV) [5, 11], and multi-modal modeling [6, 27, 39]. Demonstrating remarkable efficacy across domains, this method enhances model performance in terms of accuracy and efficiency on downstream tasks.

Subsequently, various downstream tasks involving multiple modalities are to show improved performance after fine-tuning with

---

*Corresponding author

the pre-train model. For instance, sarcasm–a prevalent and intricate mode of communication–presents a significant challenge in the realm of multi-modal learning. Accurate detection of sarcasm is vital for numerous applications, including sentiment analysis [9, 17, 51], social media monitoring [1], and conversational systems [21, 26, 43, 50]. Traditional sarcasm detection methods primarily rely on textual features, which may not capture the full spectrum of information required for accurate detection. With the increasing availability of multi-modal data, including text, images, audio, and videos, researchers have started exploring the potential of integrating these diverse data sources to enhance multi-modal sarcasm detection (MSD) performance [3, 45, 52, 59]. This advancement demands that models possess the ability to recognize the cues across modalities accurately. The complexity of the task requires more power of the multi-modal model.

Pretraining involves training a model on a large corpus of data to learn broad and generalizable representations, which can then be adapted to specific tasks, including MSD, through a process known as fine-tuning. The pretrain-finetune paradigm, involving the initial training of models on large-scale datasets to learn broad, generalizable representations followed by task-specific fine-tuning, is a common approach. Some studies in multi-modal sarcasm detection have adopted this strategy, attempting to learn the incongruity between textual and visual modalities by merging features from multiple sources. However, fully fine-tuning these models for complex tasks requires training numerous parameters, which can be resource-intensive [2, 8, 47, 48].

Therefore, finding new ways to efficiently transfer existing foundation models to downstream tasks without incurring excessive costs becomes an important challenge in the field. The Parameter Efficient Fine-tuning (PEFT) method aims to solve these problems. By fine-tuning only a select few additional model parameters and freezing the majority of the pre-trained model's parameters, PEFT significantly cuts down on computational and storage demands. This method has been demonstrated to outperform traditional fine-tuning in settings with limited data and exhibits superior generalization [23, 28, 34]. It is versatile enough to be applied across different modalities. With PEFT, only a few training weights are introduced to the top layer of the pre-trained model. This allows the same pre-trained model to be repurposed for multiple tasks by adding minimal weights, thereby eliminating the need to full fine-tune the entire model. In short, PEFT methods enable achievement of performance comparable to that of full-parameter fine-tuning while requiring only a limited number of trainable parameters. However, the effectiveness of existing PEFT methods in handling the complexities of MSD remains less than optimal.

Consequently, we introduce a novel approach known as the **Bi**directional **A**dapter (BA) and propose a plug-in, termed MoBA, which is designed for seamless integration into MSD models to significantly enhance their performance. As illustrated in Figure 1, MoBA achieves impressive results with a minimal number of training parameters. Our approach directly addresses the challenges of large-scale modeling by refining both pre-training and fine-tuning techniques to suit the complexities of MSD. We utilize BA to dynamically extract relevant features from changes in each modality. This method allows each modal branch to learn prompt information from the alternate modality and combine it with the feature information

of its own modality, thereby improving the model's representation capabilities. Furthermore, MoBA adopts a mixture-of-experts (MoE) style approach. A wealth of research and experimental evidence has shown that MoE can significantly enhance the model's capacity to tackle various downstream tasks through fine-tuning with extensive instructional data [12, 53]. Therefore, we have also leveraged MoE to optimize BA's performance. Moreover, MoBA can be appended to the pre-trained model without adding numerous trainable parameters, maintaining efficiency in model adaptation.

The main contributions of this paper are as follows: (1) To our knowledge, this is the first work to apply PEFT to the MSD task. Our innovative module is designed for plug-and-play integration into existing MSD models, significantly reducing the need for trainable parameters by freezing most of them and utilizing only a select few for fine-tuning. (2) We introduce a novel approach, termed MoBA, specifically tailored for the MSD task. This approach combines the PEFT method with a MoE strategy. It adeptly merges multi-modal information in a dynamic fashion, facilitating effective cross-modal interactions through a straightforward and efficient framework. (3) Our experimental findings highlight MoBA's exceptional performance on MSD benchmarks, showcasing significant advancements over other PEFT approaches and previous baselines. We provide a comprehensive analysis, supported by extensive experimental data, to underscore the effectiveness of our proposed method.

## 2 Related Work

### 2.1 Multi-modal Sarcasm Detection

Multi-modal sarcasm detection (MSD) is an area of research that has gained significant attention due to its importance in understanding human communication, particularly on social media platforms, where text and images are often used together to convey messages. The initial approach to the MSD task utilized both textual and visual information [41]. Subsequently, a significant effort established an MSD benchmark and introduced a hierarchical fusion model to integrate these modalities [3]. Most recently, advancements have been made by utilizing the capabilities of pre-trained CLIP [39] models to perform MSD, achieving state-of-the-art results [38].

Despite significant advancements, MSD remains a challenging task, and relies on full fine-tune for the pre-trained model, which is too resource intensive. Additionally, whether the model can truly understand the signals of sarcasm between different modalities is also open to question. In response to these challenges, we propose MoBA, a low-rank approach that utilizes bidirectional adapters for efficient interaction between the two modalities, further enhancing the sarcasm detection capabilities. This method aims to overcome the limitations of previous approaches, improving both the efficiency and effectiveness of multi-modal sarcasm detection.

### 2.2 Parameter-Efficient Fine-Tuning

Parameter-efficient fine-tuning (PEFT) has become a vital area of research in deep learning, particularly due to the increasing scale of pre-trained models. One of the initial methods to achieve this efficiency is the use of adapter modules [19]. These small, trainable layers are strategically inserted between the layers of a pre-trained model. During fine-tuning, only the parameters of these adapter

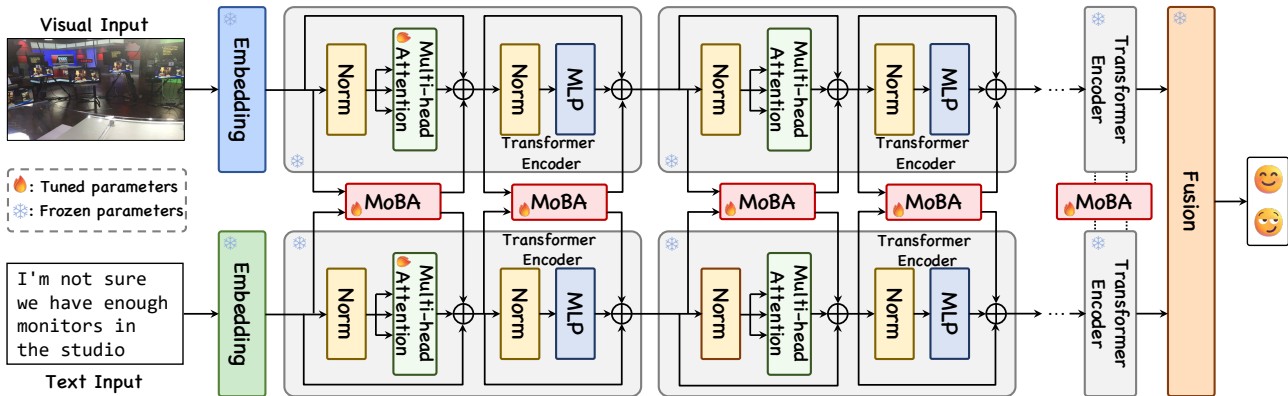

**Figure 2: Illustration of the model inserted our proposed module.**

modules are updated, leaving the original model parameters unchanged. Another approach to improving the efficiency of parameters is prompt tuning, which involves adding a few trainable tokens (prompts) to the input sequence [25]. These prompts are optimized during fine-tuning, directing the pre-trained model to produce outputs that align with the target task while maintaining fixed model parameters. Low-rank factorization techniques also contribute to parameter efficiency by applying matrix decomposition to reduce the parameter space of the neural network weights [20]. This allows for efficient fine-tuning by optimizing a smaller subset of factors rather than the entire set of weight matrices, which simplifies the model architecture without compromising performance. Despite these advances, these methods still require some retraining. As a result, we aim to explore alternative strategies for MSD that may circumvent or minimize the need for extensive retraining.

### 2.3 Mixture of Experts

The mixture-of-experts (MoE) model, first introduced by [22], represents a pioneering approach in machine learning, aiming to divide complex tasks into simpler sub-tasks that can be solved by specialized models, termed "experts". Over the years, this framework has seen significant developments and applications across various domains. With the rise of deep learning, the MoE has been particularly influential in the realm of neural networks. It addresses the challenges posed by diverse data types by utilizing contextual cues or input features to direct information to the most pertinent experts. The application of this approach has led to an improvement in the model's efficacy [7, 12, 42]. Moreover, the MoE concept has been adapted for multi-modal learning, where it adeptly processes different data sources such as text, images, and audio [4, 35]. In these contexts, MoE models promote the integration of insights from multiple modalities, thereby enhancing the predictive accuracy and coherence of the model's outputs across diverse input types.

Recently, there has been a surge in research focusing on MoE models within the context of low-rank approximations. The fusion of low-rank techniques with MoE is especially prominent in fields where models are required to handle high-dimensional data efficiently [13, 20, 49]. By enabling more streamlined and resource-efficient computations, low-rank MoE methods pave the way for

deeper and more intricate ensemble learning strategies. Overall, MoE has shown robust performance across a diverse range of tasks. In this study, we explore the application of MoE within the context of adapters for the MSD models. Our approach presents distinct advantages over the traditional, complex retraining modules employed in previous methodologies. By utilizing MoE, we provide a more efficient and effective means of enhancing model performance through low-rank techniques.

## 3 Methodology

Our proposed module can be seamlessly integrated into the MSD models, effectively freezing most of the training parameters. Within this framework, MoBA remains trainable, adept at extracting information from both modalities for bidirectional interaction. In the following, we discuss the relevant details.

### 3.1 Preliminary

For each input sample $X$ from the training dataset, there are two modalities to consider: **t**ext (T) and **v**isual (V). The inputs from these two modalities are processed through encoders to obtain their respective embeddings. These embeddings are subsequently fused to produce a predicted outcome. The process involves training a model, denoted by $f(\cdot)$, to perform the MSD task. The objective of the task is to determine whether a given sample contains sarcasm. Given the multi-modal inputs, the model predicts an outcome represented by $Y = f(T, V) \in \{0, 1\}$, where $Y = 1$ indicates the presence of sarcasm in the sample, and $Y = 0$ indicates its absence.

### 3.2 Overall Module

As shown in Figure 2, we predominantly freeze the parameters within the model. Upon being introduced to the pre-trained models, inputs from the text and visual modalities are transformed into their respective embeddings. Subsequently, we integrate our novel module into the model framework.

This module is built on the foundation of transformer-based encoder layers [46]. It is designed to deeply encode the embeddings, enhancing their representational richness. In our configuration, MoBA inputs the features before normalization and outputs at the residual addition. Each transformer layer includes two MoBAs,

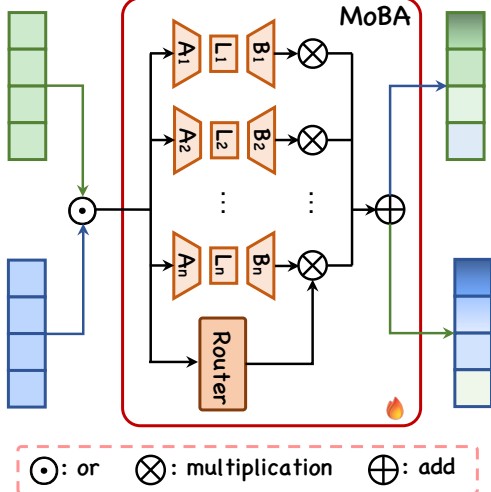

**Figure 3: The detailed architecture of MoBA. It is composed of multiple bidirectional adapters and a router that assigns the weights to these adapters. $A_i$ and $B_i$ are matrices that modify the dimensions of the embedding, whereas $L_i$ maintain a constant dimension as matrices. The operation "or" indicates that two embeddings are input separately rather than jointly.**

designed to enhance feature integration. Notably, the multi-head attention mechanisms of the transformer layers remain frozen, with the sole exception of the first layer, which is adaptable for tuning.

After obtaining the new information feature $\mathbf{E}_{i \in \{T,V\}}$, our module seamlessly integrates with existing models, facilitating effective fusion of modal information. To optimize this process, we experiment with three different fusion techniques:

(1) Concatenate fusion: This method combines features by concatenating text and visual embeddings, represented as $Y = concat(\mathbf{E}_T, \mathbf{E}_V)$. This technique is simple but effective in different scenarios;

(2) Guided weight fusion: A guided weighting mechanism calculates weights for each modality, $\alpha_i = softmax(\mathbf{W}_{i1} \cdot tanh(\mathbf{W}_{i2}) \cdot \mathbf{E}_i)$. The final output is a weighted sum, $Y = \sum \alpha_i \mathbf{E}_i$, rendering it particularly suitable for tasks where the precise modeling of the interplay between multiple modalities is crucial;

(3) Attention fusion: This approach employs an attention mechanism, where attention weights are computed as $\mathbf{W}_T, \mathbf{W}_V = softmax(\mathbf{W}(\mathbf{E}_T, \mathbf{E}_V))$, and the output is an attention-weighted combination, $Y = \mathbf{W}_T\mathbf{E}_T + \mathbf{W}_V\mathbf{E}_V$. This method significantly enhances the interactivity between modalities and is especially well-suited for contexts within multi-modal learning where such interactions are effective.

The fusion will not impact the interaction within the ABC. Furthermore, the effects of our three distinct fusions will be experimented in Table 2 to verify their influence.

Following the previous works [38], we employ the cross-entropy loss to optimize the MSD task: $\mathcal{L} = \hat{y}log(y) + (1 - y)log(1 - \hat{y})$, where $y$ is the ture label and $\hat{y}$ is our predict value.

### 3.3 MoBA

The core element of our module is the MoBA. This model is characterized by its fully trainable parameters. As depicted in Figure 3, the MoBA is meticulously crafted to ensure the smooth transition of features between different modalities. This intricate architecture is instrumental in bolstering the model's proficiency in understanding and processing multi-modal information. It is important to note that the dimensions remain unchanged before and after passing through the MoBA. To ensure that the lengths of the two modalities entering the MoBA are identical, we apply zero-padding to the modality features after they have passed through the embedding.

*Bidirectional Adapter.* Bidirectional Adapter (BA) is engineered to independently process inputs from each modality before serving as the output for another modality, thereby fostering interaction and synergy between modalities. Initially, the hidden dimension $d_e$ is reduced to $d_t$ through a down-projection layer, where $d_t$ is significantly smaller than $d_e$. This reduction is crucial for managing computational complexity. Subsequently, the embeddings pass through both a linear projection layer and an up-projection layer, which collaboratively restore the hidden dimension to $d_e$. The output from this streamlined, low-rank structure is then used to interact with features from the other modality. This configuration utilizes specialized low-rank matrices to facilitate the interactions between modalities, which are particularly skilled at detecting subtle cues that are indicative of inter-modal sarcasm. By doing so, it markedly improves the model's capacity to accurately interpret and process complex multimodal data.

*Mixture of Expert for BA.* Contrasting with the aforementioned method, MoBA introduces a more intricate structure. It enhances its design by integrating the BA within a Mixture of Experts (MoE) framework, which allows for flexibility in the number of experts across each layer. MoBA utilizes multiple sets of low-rank matrices, with each set identified as a BA expert, comprising a total of $n$ BAs throughout the entire model. The routing module, denoted as $\mathbf{G}$, is employed to direct each input token to the appropriate BA experts. This module calculates a weight vector through a linear transformation and then applies a softmax layer to distribute weights among the BA experts effectively. This dynamic weighting mechanism ensures that each input is processed by the most relevant experts based on the task.

The architecture of MoBA is meticulously crafted for adaptability, allowing for the fine-tuning of the number of BA experts to enhance performance. The incorporation of the MoE strategy within the BA framework significantly improves the model's efficacy, providing a customized approach to MSD. This sophisticated structure not only adapts to the complex demands of MSD but also optimizes the interaction and processing capabilities of the model. The formulas of the whole MoBA are as follows:

$$\mathbf{H}_{ik} = \mathbf{A}_k\mathbf{L}_k\mathbf{B}_k\mathbf{E}_i, \ i \in \{T,V\}, \ k \in \{1, 2, ..., n\} \quad (1)$$

$$\mathbf{G}_i = softmax(\mathbf{W}_i\mathbf{E}_i), \ i \in \{T, V\} \quad (2)$$

$$\mathbf{E}'_i = \sum_{k=1}^{n} \mathbf{H}_{ik}\mathbf{G}_i, \ i \in \{T, V\} \quad (3)$$

$$\mathbf{E}''_i = \mathbf{E}_i + \mathbf{E}'_j, \ i \neq j, \ i \in \{T, V\}, \ j \in \{T, V\} \quad (4)$$

Here, $\mathbf{E}$ denotes the initial embedding serving as the input, $\mathbf{E}'$ is the updated embedding following transformation and interaction, and $\mathbf{E}''$ is utilized as the final embedding for subsequent processing. The variables $\mathbf{A}$, $\mathbf{L}$, $\mathbf{B}$, and $\mathbf{W}$ represent the weight matrices within the adapter.

## 4 Experiments

To evaluate the effectiveness of our proposed module, we conducted experiments on two benchmark datasets and compared our method with state-of-the-art sarcasm detection techniques. In this section, we describe the datasets, experimental setup, results, and analysis.

### 4.1 Experiment Setup

*4.1.1 Datasets.* In our experiments, we utilized two significant datasets: MMSD [3] and MMSD2.0 [38]. MMSD dataset comprises English tweets from Twitter (recently rebranded as X), labeling positive examples by the presence of specific hashtags like #sarcasm, while tweets without such tags are classified as negative examples. MMSD2.0 dataset refines MMSD dataset by removing misleading signals and correcting inaccurate annotations, aiming for a more precise benchmark. The detailed characteristics of these datasets are presented in Table 1.

**Table 1: Statistics of two datasets.**

| Dataset | Train | Validation | Test |
|---------|-------|------------|------|
| MMSD | 39,632 | 4,820 | 4,818 |
| MMSD2.0 | 39,628 | 4,820 | 4,818 |

*4.1.2 Evaluation Metrics.* Following the previous studies [38], we adopt several key metrics to assess our model's performance: accuracy (Acc.), precision (P), recall (R), and the F1-score (F1). These metrics enable us to conduct a comprehensive and detailed evaluation, shedding light on various aspects of our model's effectiveness in accurately identifying and classifying instances in MSD task.

*4.1.3 Baselines.* To establish the efficacy of our proposed method, we conduct an extensive comparison with a diverse range of cutting-edge and traditional approaches. This comprehensive analysis is designed to provide a robust evaluation of our method's performance by benchmarking it against a variety of established baselines:

- **Text modality:** We selected influential models that have significantly advanced natural language processing. Text-CNN [24] provides a robust baseline with its ability to capture local text dependencies. Bi-LSTM [15] enhances sequence models by effectively capturing temporal dependencies through its bidirectional structure. BERT [10] revolutionizes language understanding with its deep bidirectional text representation and pre-training approach.
- **Image modality:** We choose models that have been pivotal in the advancement of computer vision. Resnet [18] overcomes the vanishing gradient problem with its deep residual network architecture, forming the basis for many subsequent innovations. ViT [11] repurposes the transformer architecture for image analysis, setting new benchmarks in image classification.

- **Multi-modal modality:** For multi-modal tasks, where the integration of different types of modality is crucial, we select a variety of models that have shown excellence in handling such complexity. HFM [3], or Hierarchical Fusion Model, is a multi-modal model that effectively combines information from different modalities through a hierarchical fusion approach. Att-BERT [36] introduces attention mechanisms to the BERT model, allowing for more focused and context-aware representations in multi-modal scenarios. InCross-MGs [29] utilizes intersection and multiplication of modality-specific graph representations to capture cross-modal dependencies. This method has shown to be effective in tasks where the relationships between modalities are complex. CMGCN [30], leverages graph convolutional networks to integrate information from different modalities in a structured manner. HKE [32] is a model that focuses on the alignment and integration of keypoints across modalities, providing a structured approach to multi-modal fusion. Lastly, Multi-view CLIP [38] extends the CLIP model to handle multiple views of data, enabling a more nuanced understanding of the relationships between text and images.

*4.1.4 Experimental Setup.* Our proposed method, alongside the baseline models for comparison, was developed using the PyTorch framework [37]. For embedding the inputs, we employed the *CLIP* as the backbone. The transformer encoder layers within our proposed model consists of 3 layers. We optimized the parameters of our model using the AdamW optimizer [33]. The training was conducted with a batch size of 64 and over 10 epochs. To ensure a fair and equitable comparison with the existing models, we meticulously followed the training configurations as described in their respective original publications. This included adhering to specified loss functions, batch sizes, and learning rate schedules, providing a consistent basis for comparison and ensuring that the evaluation of our proposed model's performance was conducted under comparable conditions. All computational experiments were conducted on a single V100 GPU.[1]

## 4.2 Main Results

The results of our experiments are presented in Table 2, which showcases the performance of the three fusion methods mentioned above plus our proposed MoBA, against the baseline model for performance. The detailed examination of the results yields several observations: (1) Multi-modal models consistently outperform the single-modal models, confirming the hypothesis that integrating information from multiple modalities typically results in more robust and accurate models. This is particularly true in complex tasks requiring nuanced understanding, such as sarcasm detection. (2) MoBA demonstrates that it is possible to achieve significant performance improvements while using considerably fewer training parameters. This not only makes the MSD models more efficient, but also reduces the computational cost associated with training. (3) Our model not only meets but surpasses the performance of existing models. Specifically, MoBA achieves superior results compared to the original state-of-the-art on the MSD dataset, recording scores of 88.40% accuracy, 82.04% precision, 88.31% recall, and 84.85% F1-score. These improvements are particularly noteworthy with our

**Table 2: Main results across different datasets. "TP" denotes the trainable parameter size. Results with † denote that we re-implemented the model. Results with bold represent our model improves over all baselines at $p < 0.05$.**

| Modality | Model | MMSD | | | | MMSD2.0 | | | |
|---|---|---|---|---|---|---|---|---|---|
| | | Acc. (%) | P (%) | R (%) | F1 (%) | Acc. (%) | P (%) | R (%) | F1 (%) |
| Text | TextCNN [24] | 80.03 | 74.29 | 76.39 | 75.32 | 71.61 | 64.62 | 75.22 | 69.52 |
| | Bi-LSTM [15] | 81.90 | 76.66 | 78.42 | 77.53 | 72.48 | 68.02 | 68.08 | 68.05 |
| | BERT† [10] (TP=110 M) | 83.60 | 78.50 | 82.51 | 80.45 | 76.52 | 74.48 | 73.09 | 73.91 |
| Image | Resnet [18] | 64.76 | 54.41 | 70.80 | 61.53 | 65.50 | 61.17 | 54.39 | 57.58 |
| | ViT† [11] (TP=86 M) | 68.51 | 57.19 | 70.83 | 63.46 | 71.80 | 64.96 | 75.15 | 69.62 |
| Text + Image | HFM [3] (TP=53 M) | 83.44 | 76.57 | 84.15 | 80.18 | 70.57 | 64.84 | 69.05 | 66.88 |
| | Att-BERT [36] (TP=136 M) | 86.05 | 80.87 | 85.08 | 82.92 | 80.03 | 76.28 | 77.82 | 77.04 |
| | InCrossMGs [29] (TP=198 M) | 86.10 | 81.38 | 84.36 | 82.84 | – | – | – | – |
| | CMGCN [30] (TP=198 M) | 86.54 | – | – | 82.73 | 79.83 | 75.82 | 78.01 | 76.90 |
| | HKE† [31] (TP=113 M) | 87.39 | 81.40 | 86.93 | 84.07 | 76.39 | 73.50 | 75.96 | 74.71 |
| | Multi-view CLIP† [38] (TP=156 M) | 88.22 | 82.03 | 88.13 | 84.97 | 85.14 | 80.33 | 88.24 | 84.09 |
| | MoBA + Fusion1 (TP=5.6 M) | 87.52 | **82.08** | 88.03 | 83.94 | 83.76 | 78.08 | **88.61** | 82.44 |
| | MoBA + Fusion2 (TP=5.6 M) | 88.07 | **82.13** | 87.85 | 84.55 | 85.01 | **80.46** | 87.67 | 83.64 |
| | MoBA + Fusion3 (TP=5.6 M) | **88.40** | 82.04 | **88.31** | 84.85 | **85.22** | 79.82 | **88.29** | **84.11** |

**Table 3: Comparison results across different datasets. "TP" denotes the trainable parameter size.**

| Modality | Model | MMSD | | | | MMSD2.0 | | | |
|---|---|---|---|---|---|---|---|---|---|
| | | Acc. (%) | P (%) | R (%) | F1 (%) | Acc. (%) | P (%) | R (%) | F1 (%) |
| Text | BERT [10] (TP=110 M) | 83.60 | 78.50 | 82.51 | 80.45 | 76.52 | 74.48 | 73.09 | 73.91 |
| | BERT + Adapter (TP=0.9 M) | 81.56 | 76.34 | 79.67 | 77.78 | 73.84 | 74.51 | 75.38 | 71.92 |
| | BERT + LoRA (TP=0.3 M) | 83.35 | 78.73 | 81.52 | 80.10 | 75.23 | 74.45 | 73.12 | 73.78 |
| Image | ViT [11] (TP=86 M) | 68.51 | 57.19 | 70.83 | 63.28 | 71.80 | 64.96 | 75.15 | 69.68 |
| | ViT+ Adapter (TP=0.9 M) | 65.23 | 55.84 | 69.12 | 61.77 | 70.56 | 64.97 | 70.34 | 67.55 |
| | ViT + LoRA (TP=0.3 M) | 67.28 | 57.12 | 70.45 | 63.09 | 70.34 | 64.67 | 74.90 | 69.41 |
| Text + Image | HKE [31] (TP=113 M) | 87.39 | 81.40 | 86.93 | 84.07 | 76.39 | 73.50 | 75.96 | 74.71 |
| | HKE + Adapter (TP=1.8 M) | 85.45 | 79.32 | 85.68 | 82.38 | 74.15 | 72.12 | 74.56 | 73.32 |
| | HKE + MoBA (TP=5.6 M) | **87.39** | **82.12** | **87.56** | **84.28** | **76.78** | **73.64** | 75.23 | 74.43 |
| | Multi-view CLIP [38] (TP=156 M) | 88.22 | 82.03 | 88.13 | 84.97 | 85.14 | 80.33 | 88.24 | 84.09 |
| | Multi-view CLIP + Adapter (TP=0.1 M) | 85.58 | 80.45 | 87.62 | 83.88 | 84.49 | 79.38 | 87.76 | 83.36 |
| | Multi-view CLIP + MoBA (TP=3.3 M) | **88.96** | **82.84** | 88.12 | **85.40** | **85.83** | 80.42 | **88.67** | **84.34** |

fusion techniques, which more effectively leverage the strengths of both textual and visual modalities. The success of our approach highlights its potential to deliver insightful and valuable outcomes through sophisticated inter-modal interactions.

## 4.3 PEFT Comparison

As shown in Table 3, we evaluate the performance of each tuning method by conducting ablation experiments on the same MSD dataset. For these evaluations, we selected some tuning methods: (1) Full-parameter fine-tuning: This traditional method updates all model parameters for the specific task, typically achieving high performance by allowing complete data fitting. However, it can be computationally expensive and risks overfitting, especially with large models and datasets. (2) Adapter tuning: Using methods described in [19] for Bert-based or/and ViT encoders, and another technique from [14] for the CLIP encoder, this strategy involves inserting a small trainable network into each layer of a pre-trained model while keeping the original parameters unchanged. This approach

improves the efficiency of the parameters and adaptability to new tasks. (3) LoRA: Described in [20], this method incorporates trainable low-rank matrices parallel to existing weights, reducing the need for extensive retraining and minimizing parameter updates while aiming to preserve performance. Although less resource-intensive, LoRA may not capture task complexity as effectively as full-parameter fine-tuning. These methods allow us to test the effectiveness of different tuning strategies on the same model.

The results indicate that while full-parameter fine-tuning often delivers the highest performance, its substantial computational demands and potential for overfitting restrict its widespread use. In contrast, the LoRA and adapter modules offer improved parameter efficiency, though they may slightly underperform in comparison. However, our proposed MoBA matches or even exceeds the performance of full-parameter fine-tuning while also maintaining computational efficiency. This is particularly evident in its ability to handle complex multi-modal data effectively. MoBA is designed to process and integrate information from various modalities using

an innovative fusion strategy coupled with a parameter-efficient fine-tuning mechanism. It significantly reduces the number of trainable parameters while preserving the model's capability to adapt to each specific modality.

## 4.4 Method Analysis

*4.4.1 The Impact of Component.* Our analysis begin by examining the role of the MoE and bidirectional adapter mechanism within the MoBA framework. The MoE approach facilitates dynamic allocation of inputs to the most appropriate adapters. In contrast, we designate a scenario where a BA is used without the MoE feature as "*w/o MoE*". Additionally, we assess the significance of the bidirectional nature of the adapters in MoBA. The BA is designed to process inputs from one modality and transfer them to another, thereby enhancing their interaction. To evaluate the impact of this feature, we modify the adapter to maintain input within its original modality after processing, termed as "*w/o Bi*".

As shown in Table 4, we compare these configurations against two baseline models – HKE and Multi-view CLIP – on the MMSD2.0 dataset. Our findings underscore the crucial contributions of both the MoE and BA to the MoBA's performance in MSD tasks. Eliminating these key components results in noticeable performance declines; for example, under the F1 metric, removing MoE leads to decreases of 1.66% and 1.94%, while omitting the bidirectional feature results in reductions of 0.81% and 3.49%, respectively. This analysis confirms the efficacy of our proposed MoBA. By integrating these essential elements, the MoBA effectively captures and processes multi-modal cues, leading to a more accurate and robust sarcasm detection model.

**Table 4: Ablation study about MoBA on MMSD2.0.**

| Model | Metric | All | *w/o* MoE | *w/o* Bi |
|---|---|---|---|---|
| HKE + MoBA | Acc. (%) | 76.78 | 75.45 | 76.15 |
| | F1 (%) | 74.43 | 72.77 | 73.62 |
| Multi-view CLIP + MoBA | Acc. (%) | 85.83 | 84.01 | 83.14 |
| | F1 (%) | 84.34 | 82.40 | 80.85 |

*4.4.2 Low-resource Settings.* To further assess the efficacy of MoBA in low-resource scenarios, we conducted a series of experiments with different amounts of training data, specifically using 10%, 20%, and 50% of the available samples, as per the methods described in [38].

As shown in Figure 4, MoBA, when integrated with Multi-view CLIP, consistently surpasses the performance of the baseline model under these constrained conditions. This can be attributed to MoBA's ability to enhance interactions within the model, thus boosting its overall capabilities. This notable improvement highlights our model's stability and reliability across varied levels of available resources, showcasing significant performance gains even with limited training data. This robustness is particularly valuable in scenarios with uneven distributions of training and testing samples, confirming MoBA's effectiveness in optimizing resource usage while maintaining high performance standards.

---

[1]https://github.com/Evfidiw/MoBA

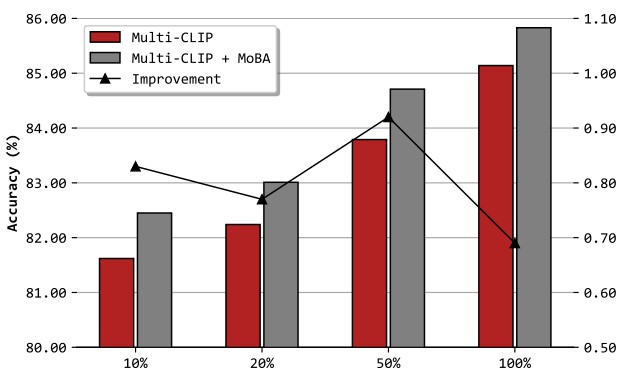

**Figure 4: In the low-resource, Multi-view CLIP incorporates MoBA in ablation experiments on MMSD2.0.**

*4.4.3 Generalizability on Sentiment.* To assess the generalizability of our proposed MoBA across different tasks, we conducted a comprehensive evaluation using multiple datasets and various sentiment classification scenarios. We utilized several well-known multi-modal sentiment analysis (MSA) datasets, including MOSI [56] and MOSEI [55]. Additionally, we tested MoBA against three established MSA models: self-MM [17], MMIM [16], and ConFEDE [54].

As detailed in Table 5, the results confirm that MoBA exhibits strong generalizability across various MSA tasks. For instance, integrating MoBA with the three aforementioned MSA models on the MOSI dataset resulted in performance improvements of 0.90%, 0.87%, and 0.78% in two-class accuracy (Acc-2) and 0.74%, 0.80%, and 1.31% in seven-class accuracy (Acc-7), respectively. Overall, this generalizability analysis for sentiment analysis reveals that MoBA, originally developed for MSD, also shows considerable promise for broader applications in other multi-modal contexts. These findings will inform further enhancements of the model, aiming to improve its performance across a broader spectrum of multi-modal learning.

**Table 5: Performance on two datasets for multi-modal sentiment analysis.**

| Model | MOSI | | MOSEI | |
|---|---|---|---|---|
| | Acc-2 (%) | Acc-7 (%) | Acc-2 (%) | Acc-7 (%) |
| self-MM | 82.33 | 82.71 | 82.49 | 83.51 |
| self-MM + MoBA | 83.23 | 83.45 | 83.75 | 83.79 |
| MMIM | 82.81 | 82.97 | 82.29 | 83.38 |
| MMIM + MoBA | 83.68 | 83.77 | 83.81 | 84.05 |
| ConFEDE | 84.43 | 84.52 | 84.48 | 84.60 |
| ConFEDE + MoBA | 85.21 | 85.83 | 85.08 | 85.16 |

*4.4.4 Sensitivity of Hyper-parameters.* In our analysis, we explore the sensitivity of crucial hyperparameters within our model, specifically the number of experts in the MoE and the number of layers in the transformer architecture. These factors significantly impact the model's complexity, computational efficiency, and overall performance in the MSD task. To explore the effects of each hyperparameter, we held all other variables constant and varied only one

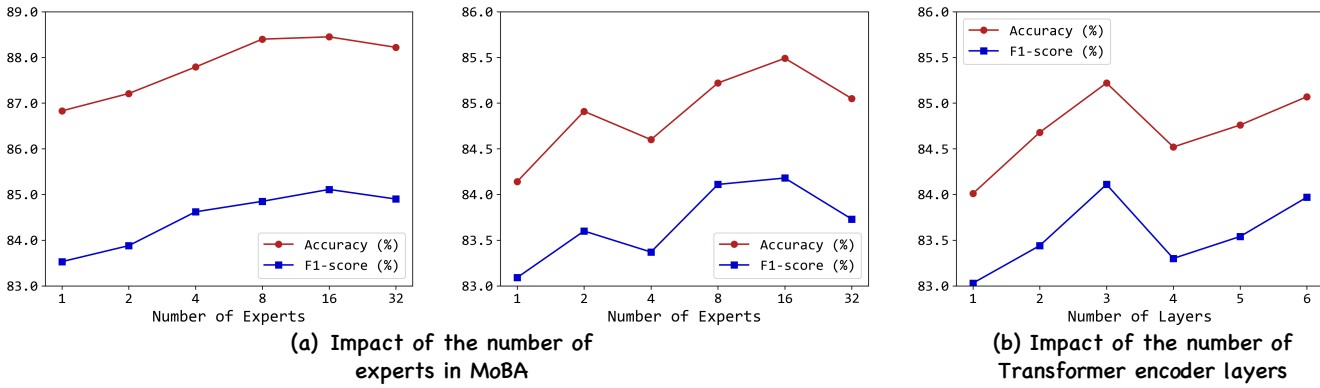

Figure 5: Performance varies with changes in hyper-parameters: (a) represents variations in the number of experts in MoBA, and (b) reflects changes in the number of transformer encoder layers.

hyperparameter at a time during our experiments. The results are detailed in Figure 5. The number of experts in the MoE setup is vital for the model's ability to specialize and manage diverse data effectively. Increasing the number of experts typically enhances performance; however, excessively high numbers can lead to over-fitting and unnecessarily high computational costs. In contrast, too few experts might not sufficiently address the complexity needed for effective decision-making across different inputs. Thus, identifying an optimal number of experts is crucial to balance model complexity with efficiency. Similarly, the depth of the transformer, indicated by its number of layers, profoundly influences the model's capacity to understand and process input features. More layers generally provide a deeper and more nuanced comprehension, which is advantageous for complex tasks that require detailed contextual and relational analysis, such as sarcasm detection. Nevertheless, more layers also mean more parameters, longer training times, and an increased risk of overfitting, especially with limited data. Therefore, it is important to carefully adjust the number of layers to maintain a balance between depth and operational efficiency.

Our findings indicate that there is a critical threshold where further increasing the number of experts or layers does not yield substantial performance gains and may even harm performance due to the aforementioned issues of complexity and overfitting. These insights are essential for optimizing our model to ensure it is not only effective but also resource-efficient.

*4.4.5 Error Analysis.* Figure 6 presents an error analysis of our experimental findings. This analysis indicates that a considerable number of errors stem from samples with significant textual content within the images. For example, the image in example (a) is purely textual, which challenges our model due to its lack of mechanisms for directly handling the interplay between text and image modalities. To mitigate this, we performed an additional experiment incorporating Optical Character Recognition (OCR) results into our model. The experiment results show that there has been improvements in performance. Additionally, the image in example (b) includes not only crucial textual information, but also critical expressive details from the individual depicted. This complexity underscores the need to better capture the subtle interplay between visual expressions and textual information. We investigated

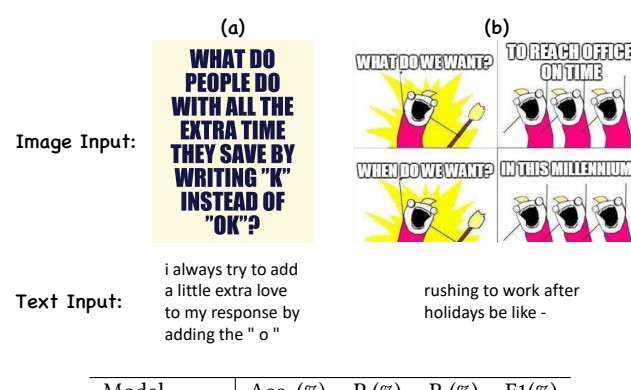

| Model | Acc. (%) | P (%) | R (%) | F1(%) |
|---|---|---|---|---|
| Ours | 85.22 | 79.82 | 88.29 | 84.11 |
| Ours + OCR | 85.34 | 78.94 | 89.65 | 84.16 |

Figure 6: Example of error analysis (Top) and experiments on incorporating OCR information (Bottom).

methods to refine the model's ability to interpret these complex interactions, aiming to enhance its capacity to identify underlying sarcastic nuances. Improving this capability could significantly advance the model's effectiveness in recognizing sarcasm across diverse modalities.

## 5 Conclusion

In this paper, we presented a novel approach for multi-modal sarcasm detection that combines the low-rank and MoE design concepts. This innovative strategy boosts the performance of models engaged in multi-modal sarcasm detection by effectively combining the strengths of both MoE and BA to facilitate dynamic inter-modal interactions. Our experimental findings consistently show that the MoBA enhances the model's accuracy and robustness while also preserving computational efficiency. MoBA adeptly captures and integrates complex inter-modal interactions, resulting in substantial improvements in sarcasm detection across a variety of datasets Moreover, the generalization and scalability of MoBA indicate its potential applicability in other areas of multi-modal learning.

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
