# OpenReview forum: "MoBA: Mixture of Bi-directional Adapter for Multi-modal Sarcasm Detection"
_acmmm.org/ACMMM/2024/Conference — MM2024 Poster_

### Official Review · Reviewer_usLt · 2024-04-28

**Rating:** 2
**Confidence:** 4

**Summary:**

The paper presents MoBA, a novel Bidirectional Adapter approach aimed at improving multi-modal sarcasm detection (MSD) involving textual and visual data. By enhancing intermodality interaction through a low-rank representation, MoBA efficiently reduces training parameters to about 3.29 million and significantly boosts sarcasm detection accuracy. Empirical studies on two public datasets confirm MoBA's effectiveness, highlighting its potential as a reliable and computationally efficient solution for complex multi-modal tasks.

**Strengths:**

1. This paper is well-written and well-organized.
2. MoBA significantly enhances multi-modal sarcasm detection by improving intermodality interaction with a low-rank representation, resulting in both increased accuracy and reduced computational resource requirements.

**Limitations:**

1. The authors state that the devised MoBA achieves impressive results with a minimal number of training parameters. Authors should compare computational complexity with baselines and some variants with their base model, such as finetuning CLIP.
2. Is there a large gap between MoBA and full fine-tuning?
3. Some important baselines are overlooked, such as A Semantic Enhancement Framework for Multimodal Sarcasm Detection, Debiasing Multimodal Sarcasm Detection with Contrastive Learning, and Mutual-enhanced Incongruity Learning Network for Multi-modal Sarcasm Detection.
4. Typo is existed in line 545. There should be a space between “layers” and “We”. “The transformer encoder layers within our proposed model consists of 3 layers.We optimized the parameters of our model using the AdamW optimizer”
5. CCS CONCEPTS is not correct.

**Suitability:**

3

---

### Official Review · Reviewer_S7wp · 2024-05-25

**Rating:** 4
**Confidence:** 3

**Summary:**

This paper introduces a novel multimodal sarcasm detection method, named MoBA (Mixture of Bidirectional Adapter). The proposed approach aims to enhance both the efficiency and accuracy of multimodal sarcasm detection (MSD) tasks. By integrating low-rank representation and a mixture of experts (MoE) style, MoBA stands out in its ability to minimize training parameters while maintaining high performance.

**Strengths:**

Innovation in PEFT Application: MoBA represents a pioneering effort in applying Parametric Efficient Fine-tuning (PEFT) techniques to the domain of MSD. This is significant as it addresses a crucial need to reduce the number of training parameters by freezing the majority of pre-trained model parameters and fine-tuning only a select few. This not only makes the model more efficient but also less resource-intensive.

State-of-the-Art Performance: Experimental results demonstrate that MoBA achieves state-of-the-art performance across multiple benchmarks. The integration of bidirectional adapters and the mixture of experts allows for more effective cross-modal interactions, which is pivotal in detecting sarcasm that often relies on nuanced and context-dependent cues from both text and images.

Efficiency and Scalability: By utilizing a low-rank adaptation strategy, MoBA significantly cuts down on the computational load and storage requirements. The model’s design ensures scalability, making it applicable to a wide range of tasks beyond sarcasm detection, as evidenced by its performance in other multimodal contexts.

**Limitations:**

Training Time Discussion Needed: While the focus on PEFT is commendable, the paper would benefit from a more detailed discussion on the training time implications. PEFT techniques generally aim to reduce computational overhead, but an explicit comparison of training times between MoBA and traditional full-fine-tuning methods would provide a clearer picture of its efficiency gains.

Complexity of Bidirectional Interaction: The paper introduces a sophisticated mechanism for bidirectional interaction between modalities, which, while innovative, could potentially add complexity to the implementation. Further analysis of this aspect in practical deployment scenarios would be beneficial.Definitely suitable

**Suitability:**

3

---

### Official Review · Reviewer_zauL · 2024-05-25

**Rating:** 4
**Confidence:** 2

**Summary:**

This paper directly applies LoRA struggles to fully address the intricate demands of multi-modal sarcasm detection task.

**Strengths:**

This paper captures the nuances of sarcastic expressions with a reduced number of training parameters.

**Limitations:**

1) The comparison of the basic LORA and the proposed MoBA is missing. The effectiveness cannot be confirmed.
2) The publicly Twitter Multi-Modal Sarcasm dateset should be conducted for evaluate the results comprehensively.

**Suitability:**

3

---

### Official Review · Reviewer_2AtH · 2024-06-04

**Rating:** 4
**Confidence:** 3

**Summary:**

This paper proposes a novel approach called MoBA (Mixture of Bidirectional Adapter) for multi-modal sarcasm detection (MSD). The key points are:

- MoBA combines parameter-efficient fine-tuning (PEFT) techniques with a mixture-of-experts (MoE) framework to efficiently adapt pre-trained models for MSD while reducing training parameters.
- It introduces a Bidirectional Adapter (BA) that facilitates interaction and information exchange between textual and visual modalities through low-rank representations. This allows the model to better capture sarcasm cues across modalities.
- MoBA is designed as a plug-in module that can be integrated into existing MSD models. It significantly reduces trainable parameters by freezing most of the pre-trained model and only fine-tuning a few.
- Extensive experiments on two MSD benchmark datasets show that MoBA substantially improves sarcasm detection accuracy while using much fewer training parameters compared to other state-of-the-art methods.
- Ablation studies confirm the effectiveness of the MoE and BA components in MoBA. The approach also demonstrates strong performance in low-resource settings and good generalizability to multi-modal sentiment analysis tasks.

**Strengths:**

Overall, the technical aspects of the paper seems to be sound. The introduction of the bidirectional adapter in the form of the MoBa is moderately novel, and from the experimental results do seem to achieve the goals for which it was designed. The use of a MoE setup is a good idea, since this enables cross-referencing of different perspectives which will inevitably result in the case of multimodal inputs, as is the case here. Numerical experiments seems to be quite impressive, as they suggest superior performance compared to some large models such as BERT.

**Limitations:**

On the other hand, the idea of PEFT seems to be quite ad-hoc, and is a standard technique for reducing the complexity of training a deep model; this does not seem like a significant contribution (or should not be designated as such). MoBa, although a nice design which should facilitate its integration into other models, seems to me like a repackaging of modality alignment modules in other multimodal setups. At least, I am not sure how MoBA operates differently from such alignment modules. In addition, it is unclear to me what 'FusionN', where N=1,2,3, means in the Main Results section. As I understand it, MoBA is a standalone module, which needs to be integrated with another deep model for downstream tasks. Hence it is important to state clearly what these fusion models are.

**Suitability:**

3

---

### Meta-Review · Area_Chair_TeWF · 2024-07-03

**Recommendation:** Accept (Poster)
**Confidence:** 4

**Metareview:**

This paper proposes a Bidirectional Adapter approach, with the goal of minimizing the number of parameters of models addressing the task of multi-modal sarcasm detection.

Regarding this paper's strengths, reviewers acknowledge the effectiveness of the proposed approach (2AtH, S7wp, usLt), the extensive experiments (2AtH), and that it is well-written (usLt).

Among its weaknesses, some reviewers highlight the reduced significance of the contribution (2AtH) and insufficient experimental comparisons, in particular with full model fine-tuning (S7wp, usLt).
Reviewers also mentioned a weak discussion of the implications of attaching MoBA to other models/downstream tasks (2AtH, S7wp).

The authors' rebuttal addressed some of the reviewers' concerns, providing new results to further strengthen experimental evaluation.

Therefore, to sum up, this paper proposes an interesting and relevant approach but in its current form, experimental evaluation missed some critical aspects and in-depth discussions. Therefore, I suggest this work be accepted if there is space, provided that authors accommodate the clarifications and results provided in the rebuttal, in the paper final version.